



# Feature scale and identifiability: How much information do point hydraulic measurements provide about heterogeneous head and conductivity fields?

Scott K. Hansen[1], Daniel O'Malley[2], and James P. Hambleton[3]

[1]Zuckerberg Institute for Water Research, Ben-Gurion University of the Negev, Israel
[2]EES-16, Los Alamos National Laboratory, USA
[3]Department of Engineering, Cambridge University, UK

**Correspondence:** Scott K. Hansen (skh@bgu.ac.il)

**Abstract.** We systematically investigate how the spacing and type of point measurements impacts the scale of subsurface features that can be identified by groundwater flow model calibration. To this end, we consider the optimal inference of spatially heterogeneous hydraulic conductivity and head fields based on three kinds of point measurements that may be available at monitoring wells: of head, permeability, and groundwater speed. We develop a general, zonation-free technique for Monte Carlo (MC) study of field recovery problems, based on Karhunen-Loève (K-L) expansions of the unknown fields whose coefficients are recovered by an analytical, continuous adjoint-state technique. This allows unbiased sampling from the space of all possible fields with a given correlation structure and efficient, automated gradient-descent calibration. The K-L basis functions have a straightforward notion of period, revealing the relationship between feature scale and reconstruction fidelity, and they have an *a priori* known spectrum, allowing for a non-subjective regularization term to be defined. We perform automated MC calibration on over 1100 conductivity-head field pairs, employing a variety of point measurement geometries and evaluating the mean-squared field reconstruction accuracy, both globally and as a function of feature scale. We present heuristics for feature scale identification, examine global reconstruction error, and explore the value added by both the groundwater speed measurements and by two different types of regularization. We find that significant feature identification becomes possible as feature scale exceeds four times measurement spacing and identification reliability subsequently improves in a power law fashion with increasing feature scale.

## 1 Introduction

### 1.1 Motivation

For many forms of groundwater modeling, it is common to infer the spatially heterogeneous hydraulic conductivity and head fields which together determine groundwater flow from sparse, passively collected measurements at groundwater wells. While significant high-resolution information about heterogeneity may be obtained from a wide variety of interventions including geophysical measurements (Slater, 2007; Szabó, 2015), hydraulic tomography (Illman et al., 2007) and both push-pull (Hansen et al., 2017) and point-to-point (Irving and Singha, 2010) tracer tests, such data is often unavailable at scales of interest. Due





to their prevalence in practical, field-scale model calibration, we focus here on three types of point information that are readily and commonly determined at monitoring wells. These are hydraulic head, $\tilde{h}$, obtained from direct piezometry, Darcy flux magnitude, $\tilde{q}$, obtained from point dilution tests and core sample porosity estimates, and log hydraulic conductivity, $\ln \tilde{K}$, from slug tests or Kozeny–Carman core sample analyses. Calibrating a spatially-nonuniform hydraulic conductivity field at high resolution from sparse measurements of these three types is generally a highly under-determined inverse problem that defies exact solution.

Model calibration is premised on the assumption that fitting the observable data will sufficiently constrain unobserved underlying fields to enable useful conclusions, even given major uncertainty. However, given only point measurements of a given resolution, it is not immediately clear how accurate the estimated head and log hydraulic conductivity fields will be. Recalling the well-known 1D result that the drop in head along a streamline segment is determined by the harmonic mean of the conductivities encountered, and not their order, we see it is not possible to recover information about small-scale variations from endpoint head, flux and conductivity. However, we may gain hope from knowledge that the head field is generally much smoother than the underlying conductivity field, and that a full suite of measurements at each well will provide us with both the head, $\tilde{h}$, and $\nabla \tilde{h}$: the first two terms of the Taylor expansion of this smooth function around each well. To the extent that it is possible to correctly reconstruct the head distribution at locations away from the wells, we *will* be able to partially constrain the conductivity and to identify its large-scale features—meaning contiguous regions of similar hydraulic conductivity.

Additionally, it may be possible to employ regularization techniques that do not depend on detailed site knowledge to further improve the reconstruction quality. The question is: how much can we learn, at which scales?

To our knowledge, the degree to which density and type of measurement relate to the scale of feature that can reliably be recovered, and to the global error of the recovered field, has not previously been examined systematically. We seek to address these matters in two ways. Firstly, we aim to derive a quantitative cutoff and scaling relationship relating feature scale identifiability to measurement density. Secondly, we wish to make concrete, empirically grounded statements about the relative value of the different types of point data and regularization schemes to accurate inference of hydraulic conductivity and head fields using standard gradient-descent model calibration.

## 1.2 Previous contributions

Being essential to reliable modeling, the inverse problem of reconstructing hydraulic conductivity fields from incomplete data has long interested subsurface hydrologists. The literature on this problem has become so large over the decades that it even contains a substantial *meta*-literature of review papers (e.g., Bagtzoglou and Atmadja, 2005; Carrera et al., 2005; Franssen et al., 2009; Zhou et al., 2014). Interested readers are referred to these papers; we will only briefly summarize some key developments that contextualize our work.

From a purely mathematical standpoint, the hydraulic problem is ill-posed, being non-unique, and additional constraints are required to regularize to a unique, physically plausible solution. Neuman (1973) proposed an explicit regularization scheme employing explicit smoothness and boundedness constraints on the conductivity field. Yoon and Yeh (1976) proposed a similar scheme, in which the solution was expressed as a superposition of finite element shape functions, with the boundedness con-



straints applied to the shape functions themselves. An alternative dimension-reduction approach based on zonation with fixed parameters in each zone was proposed by Yeh and Yoon (1981). An alternative two-stage approach, limited to small conductivity variances (Zhou et al., 2014), is based on first identifying the geostatistical spatial correlations of head and permeability,

followed by co-kriging to estimate conductivity at locations of interest (Kitanidis and Vomvoris, 1983; Hoeksema and Kitanidis, 1984). In the context of the general nonlinear conductivity calibration problem, where variance is not generally small and an initial guess may be far from the true solution, iterative variants on the geostatistical approach have been proposed Kitanidis (1995); Yeh et al. (1995); Cardiff et al. (2009). Explicitly Bayesian approaches that view model error as the key source of uncertainty (Carrera and Neuman, 1986a), and in which regularization is viewed as coming from a prior probability distribution

(Kitanidis, 1986; Woodbury and Ulrych, 2000) have also been proposed. Analytical covariance relationships between head and permeability are valid for small perturbations. For work in large-dimensional spaces (such as calibration of high-resolution conductivity fields), reduction of non-uniqueness by use of principal component dimension reduction has also been proposed (Tonkin and Doherty, 2005; Kitanidis and Lee, 2014), representing a sort of pre-regularization. The relative performance of the various techniques has not much been much studied, save in the large-scale numerical study of Franssen et al. (2009) and

the bench-scale study of Illman et al. (2010), which only compared hydraulic tomography with simple averaging and kriging. Franssen et al. (2009) reported little performance difference among the various techniques they compared.

Additionally, researchers have tried approaches that improve uniqueness by addition of other forms of physical data. These include hydraulic tomography, in which the head field is manipulated by pumping at wells surrounding the area of interest (e.g., Gottlieb and Dietrich, 1995; Yeh and Liu, 2000; Illman et al., 2007), use of transient head data (e.g., Carrera and Neuman,

1986a; Zhu and Yeh, 2005), flux measurements (Tso et al., 2016), chemical (Wagner, 1992; Michalak and Kitanidis, 2004; Xu and Gómez-Hernández, 2018; Delay et al., 2019) and thermal (Woodbury et al., 1987) tracer data. These are largely out of scope for the present work, which focuses on steady-state hydraulic inversion only, although we will re-examine use of local groundwater speed (i.e., flux) information.

As inversion is typically performed via gradient descent, it is necessary to estimate the gradient of the loss function with

respect to the parameters. Where the loss function is expensive to compute and the parameter space is high dimensional, it is generally not feasible to estimate the gradient naively via successive parameter perturbation. Various solutions exist, including automatic differentiation Elizondo et al. (2002); Sambridge et al. (2007); Wu et al. (2023), and the adjoint state method, initially applied to the groundwater inverse problem by Sykes et al. (1985). Adjoint state formulations for groundwater model inversion commonly incorporate head data only (Sykes et al., 1985; Carrera and Neuman, 1986b; Lu and Vesselinov, 2015;

Delay et al., 2019), although they have been formulated to incorporate other forms of information (Cirpka and Kitanidis, 2001). Both discrete and continuous adjoint-state formulations have been studied by Delay et al. (2017) and Hayek et al. (2019), with the former identifying advantages for the continuous approach. We have not seen formulations for a loss function that includes conductivity and speed measurements, or regularization.





### 1.3 Concrete goals and overview of contents

To study the inter-relation of measurement density, feature scale and identifiability, we develop a automated gradient descent calibration Monte Carlo (MC) framework with the following properties:

1. Samples initial guesses in an unbiased fashion from the space of possible conductivity fields.

2. Avoids arbitrary *a priori* zonation and structural constraints.

3. Output easily interpreted in terms of feature scale reconstruction reliability.

4. Sufficiently low computational cost.

We satisfy the first three properties by generating fields with Karhunen-Loéve (K-L) expansions and working directly with their coefficients rather than zonating the domain and calibrating constant conductivities on fixed regions. We select a covariance kernel such that the K-L basis functions can be determined analytically, and such that they each have an obvious frequency-domain interpretation. As the basis functions are orthonormal, they admit a Fourier-type analysis of the reconstruction. The last
property is satisfied deriving a continuous adjoint-state sensitivity of loss function to measurements, reducing the computational cost to the size of the measurement vector rather than the vector of K-L coefficients.

In Section 2 we derive the continuous adjoint-state form of the optimization equations for the steady-state groundwater flow equation subject to a loss function containing point data about head, flux, and conductivity, as well as an arbitrary regularization term. In Section 3 we derive the K-L basis functions and eigenvalues, present fitness metrics, discuss regularization, and
formalize two MC studies. In Section 4 we present results, derive a quantitative relation between measurement density and feature reconstruction reliability, discuss global error, and numerically compare the utility of various types of measurements and regularization approaches. In Section 5, we summarize our key conclusions and point towards future work.

## 2 Derivation of continuous adjoint-state optimization equations

We look to solve for a spatially-distributed hydraulic conductivity field, $K(\mathbf{x}; \mathbf{p})$, which is defined by some parameter vector,
$\mathbf{p}$, and exists throughout some domain, $\Omega$, and where $\mathbf{x}$ represents the spatial coordinate within $\Omega$. Optionally, we try to solve simultaneously for the specified-head boundary condition $h|_{\partial\Omega}$. We take for granted that our system obeys the steady-state groundwater flow equation $\nabla \cdot (K \nabla h) = 0$, where $h(\mathbf{x})$ represents hydraulic head and $K$, a scalar, represents an isotropic, local hydraulic conductivity. We further assume that we have accurate measurements of head, $\tilde{h}(\mathbf{x}_j)$, and/or Darcy flux magnitude, $\tilde{q}(\mathbf{x}_k)$, and log hydraulic conductivity, $\ln \tilde{K}_l$ at a number of discrete points, $\mathbf{x}_k$, within $\Omega$.
We will be obtaining $K$ by gradient descent on some loss function, $J$, representing mismatch between model predictions and point measurements, as a function of a vector of model parameters, $\mathbf{p}$, meaning we require the sensitivity vector $\frac{\partial J}{\partial \mathbf{p}}$. However, because $h$ is measured and will appear explicitly in the formula for $J$ we expect that we will need to determine $\frac{\partial h}{\partial \mathbf{p}}$, which is computationally intractable for large-dimensional $\mathbf{p}$. Consequently, we aim derive an adjoint equation that, when satisfied, eliminates the dependence of $\frac{\partial J}{\partial \mathbf{p}}$ on $\frac{\partial h}{\partial \mathbf{p}}$. We do this below.





We begin with the Dirichlet BVP for the steady-state groundwater flow equation:

$$\nabla \cdot (K\nabla h) \quad = \quad 0 \text{ in } \Omega, \tag{1}$$

$$h \quad = \quad g(\mathbf{x}) \text{ on } \delta\Omega, \tag{2}$$

where $g(\mathbf{x})$ is a known function.

### 2.1   Loss function

We attempt to minimize $J$, which consists of as many as four metrics:

1. the weighted square error of our modeled Darcy speeds, $K(\mathbf{x}_j; \mathbf{p}) \|\nabla h(\mathbf{x}_j; \mathbf{p})\|$, relative to measurements, $\tilde{q}(\mathbf{x}_k)$ at the measurement locations, $\mathbf{x}_j$

2. the weighted squared error of our modeled heads, $h(\mathbf{x}_k)$, relative to our measurements, $\tilde{h}(\mathbf{x}_k)$, at the measurement locations, $\mathbf{x}_k$,

3. the weighted squared error of our modeled log hydraulic conductivity, $\ln K(\mathbf{x}_l)$, relative to our measurements, $\ln \tilde{K}(\mathbf{x}_k)$, at the measurement locations, $\mathbf{x}_l$, and optionally

4. a regularization term, $R$, based on prior notions of how plausible a given underlying log-conductivity is.

This can be expressed mathematically as

$$
\begin{aligned}
J \equiv & \sum_j w_j \left( K(\mathbf{x}_j; \mathbf{p}) \|\nabla h(\mathbf{x}_j; \mathbf{p})\| - \tilde{q}(\mathbf{x}_j) \right)^2 \\
& + \sum_k w_k \left( h(\mathbf{x}_k; \mathbf{p}) - \tilde{h}(\mathbf{x}_k) \right)^2 \\
& + \sum_l w_l \left( \ln K(\mathbf{x}_l; \mathbf{p}) - \ln \tilde{K}(\mathbf{x}_l) \right)^2 \\
& + R\left( K(\mathbf{x}; \mathbf{p}) \right)
\end{aligned} \tag{3}
$$

Here we explicitly acknowledge the dependence of $h$ and $K$ on a vector of model parameters, $\mathbf{p}$, which generally encodes the model parameters and the boundary conditions. The weights $w_j$, $w_k$, and $w_l$, which may be zero, may reflect the relative areas of the Voronoi regions associated with each well, as well as a relative importance rating of the types of measurement. The magnitude of the regularization function may also vary by an arbitrary scalar to change its weight relative to measurement misfit. For simplicity we will generally not write this dependence explicitly. The sensitivity of $J$ to any given model parameter,





$p_i$, can be computed directly:

$$\frac{\partial J}{\partial p_i} = \sum_j 2 w_j \left( K(\mathbf{x}_j) \| \nabla h(\mathbf{x}_j)\| - \tilde{q}(\mathbf{x}_j) \right) \left[ \frac{\partial K}{\partial p_i} \| \nabla h(\mathbf{x}_j)\| + K \frac{\nabla h \cdot \nabla \phi_i}{\| \nabla h \|} \right]$$

$$+ \sum_k 2 w_k \phi_i(\mathbf{x}_k) \left( h(\mathbf{x}_k) - \tilde{h}(\mathbf{x}_k) \right)$$

$$+ \sum_l 2 w_l \frac{1}{K(\mathbf{x}_l)} \frac{\partial K}{\partial p_i} \left( \ln K(\mathbf{x}_l) - \ln \tilde{K}(\mathbf{x}_l) \right)$$

$$+ \frac{\partial R}{\partial p_i}, \tag{4}$$

where we define the sensitivity $\phi_i \equiv \partial h / \partial p_i$. Employing vector form and rewriting in terms of integrals over the volume $\Omega$:

$$\frac{\partial J}{\partial \mathbf{p}} = \int_\Omega 2 \left( K \| \nabla h\| - \tilde{q} \right) \frac{\partial K}{\partial \mathbf{p}} \| \nabla h \| \sum_j w_j \delta(\mathbf{x} - \mathbf{x}_j) \, dV$$

$$+ \int_\Omega 2 \left( K \| \nabla h\| - \tilde{q} \right) K \frac{\nabla h \cdot \nabla \phi}{\| \nabla h \|} \sum_j w_j \delta(\mathbf{x} - \mathbf{x}_j) \, dV$$

$$+ \int_\Omega 2 \phi \left( h - \tilde{h} \right) \sum_k w_k \delta(\mathbf{x} - \mathbf{x}_k) \, dV$$

$$+ \int_\Omega 2 \frac{1}{K} \frac{\partial K}{\partial \mathbf{p}} \left( \ln K - \ln \tilde{K} \right) \sum_l w_l \delta(\mathbf{x} - \mathbf{x}_l) \, dV$$

$$+ \frac{\partial R}{\partial \mathbf{p}}. \tag{5}$$

Simplifying the second integral by employing integration by parts to remove the dependence on $\nabla \phi$ and then employing (1)

yields

$$\frac{\partial J}{\partial \mathbf{p}} = \int_\Omega 2 \left( K \| \nabla h\| - \tilde{q} \right) \frac{\partial K}{\partial \mathbf{p}} \| \nabla h \| \sum_j w_j \delta(\mathbf{x} - \mathbf{x}_j) \, dV$$

$$- \int_\Omega 2 \phi K \nabla h \cdot \nabla \left[ \left( K - \frac{\tilde{q}}{\| \nabla h \|} \right) \sum_j w_j \delta(\mathbf{x} - \mathbf{x}_j) \right] dV$$

$$+ \int_\Omega 2 \phi \left( h - \tilde{h} \right) \sum_k w_k \delta(\mathbf{x} - \mathbf{x}_k) \, dV$$

$$+ \int_\Omega 2 \frac{1}{K} \frac{\partial K}{\partial \mathbf{p}} \left( \ln K - \ln \tilde{K} \right) \sum_l w_l \delta(\mathbf{x} - \mathbf{x}_l) \, dV$$

$$+ \frac{\partial R}{\partial \mathbf{p}}. \tag{6}$$

.



## 2.2 Weak-form equation for sensitivity

We may differentiate equations (1-2) with respect to $\mathbf{p}$ so as to generate a BVP for the sensitivity vector, $\phi$:

$$\nabla \cdot \left( \frac{\partial K}{\partial \mathbf{p}} \nabla h \right) + \nabla \cdot (K \nabla \phi) \quad = \quad 0 \text{ in } \Omega, \tag{7}$$

$$\phi \quad = \quad \frac{\partial f}{\partial \mathbf{p}} \text{ on } \delta\Omega. \tag{8}$$

Sensitivity equation (7) can be placed in weak form by expressing it as an integral with respect to $\phi^*$, an arbitrary, smooth test function known as the *adjoint*:

$$\int_\Omega \left[ \nabla \cdot \left( \frac{\partial K}{\partial \mathbf{p}} \nabla h \right) \right] \phi^* \, dV + \int_\Omega \left[ \nabla \cdot (K \nabla \phi) \right] \phi^* \, dV = 0. \tag{9}$$

We then integrate the first integral by parts once and the second by parts twice, to arrive at the five-term expression

$$0 = \int_{\delta\Omega} \frac{\partial K}{\partial \mathbf{p}} \nabla h \phi^* \cdot dS - \int_\Omega \frac{\partial K}{\partial \mathbf{p}} \nabla h \cdot \nabla \phi^* \, dV$$

$$+ \int_{\delta\Omega} K \nabla \phi \phi^* \cdot dS - \int_{\delta\Omega} K \phi \nabla \phi^* \cdot dS + \int_\Omega \phi \nabla \cdot (K \nabla \phi^*) \, dV. \tag{10}$$

## 2.3 Removing derivatives of $\phi$ and simplifying

Our goal is to derive an equation for $\frac{\partial J}{\partial \mathbf{p}}$ that does not explicitly depend on $\phi$. To this end, we add (6) and (10) to generate

$$\frac{\partial J}{\partial \mathbf{p}} = \int_\Omega 2 \left( K \|\nabla h\| - \tilde{q} \right) \frac{\partial K}{\partial \mathbf{p}} \|\nabla h\| \sum_j w_j \delta(\mathbf{x} - \mathbf{x}_j) \, dV$$

$$- \int_\Omega 2\phi K \nabla h \cdot \nabla \left[ \left( K - \frac{\tilde{q}}{\|\nabla h\|} \right) \sum_j w_j \delta(\mathbf{x} - \mathbf{x}_j) \right] dV$$

$$+ \int_\Omega 2\phi \left( h - \tilde{h} \right) \sum_k w_k \delta(\mathbf{x} - \mathbf{x}_k) \, dV$$

$$+ \int_\Omega \frac{2}{K} \frac{\partial K}{\partial \mathbf{p}} \left( \ln K - \ln \tilde{K} \right) \sum_l w_l \delta(\mathbf{x} - \mathbf{x}_l) \, dV + \frac{\partial R}{\partial \mathbf{p}}$$

$$+ \int_{\delta\Omega} \frac{\partial K}{\partial \mathbf{p}} \nabla h \phi^* \cdot dS - \int_\Omega \frac{\partial K}{\partial \mathbf{p}} \nabla h \cdot \nabla \phi^* \, dV$$

$$+ \int_{\delta\Omega} K \nabla \phi \phi^* \cdot dS - \int_{\delta\Omega} K \phi \nabla \phi^* \cdot dS + \int_\Omega \phi \nabla \cdot (K \nabla \phi^*) \, dV. \tag{11}$$



We ultimately want to remove the dependence on $\phi$, so we combine the volume integrals whose integrands are proportional to it:

$$
\begin{aligned}
\frac{\partial J}{\partial \mathbf{p}} =& \int_\Omega \phi \left[ \nabla \cdot (K \nabla \phi^*) - 2K \nabla h \cdot \nabla \left( \left( K - \frac{\tilde{q}}{\|\nabla h\|} \right) \sum_j w_j \delta(\mathbf{x} - \mathbf{x}_j) \right) + 2\left( h - \tilde{h} \right) \sum_k w_k \delta(\mathbf{x} - \mathbf{x}_k) \right] dV \\
&+ \int_\Omega 2\left( K \|\nabla h\| - \tilde{q} \right) \frac{\partial K}{\partial \mathbf{p}} \|\nabla h\| \sum_j w_j \delta(\mathbf{x} - \mathbf{x}_j)\, dV \\
&+ \int_\Omega \frac{2}{K} \frac{\partial K}{\partial \mathbf{p}} \left( \ln K - \ln \tilde{K} \right) \sum_l w_l \delta(\mathbf{x} - \mathbf{x}_l)\, dV + \frac{\partial R}{\partial \mathbf{p}} \\
&- \int_\Omega \frac{\partial K}{\partial \mathbf{p}} \nabla h \cdot \nabla \phi^*\, dV + \int_{\delta\Omega} \frac{\partial K}{\partial \mathbf{p}} \nabla h \phi^* \cdot dS + \int_{\delta\Omega} K \nabla \phi \phi^* \cdot dS - \int_{\delta\Omega} K \phi \nabla \phi^* \cdot dS.
\end{aligned}
\tag{12}
$$

The first integral (and its dependence on the $\phi$) is removed if the expression in the square brackets is zero, implying that

$$
\int_\Omega \nabla \cdot (K \nabla \phi^*) - 2K \nabla h \cdot \nabla \left( \left( K - \frac{\tilde{q}}{\|\nabla h\|} \right) \sum_j w_j \delta(\mathbf{x} - \mathbf{x}_j) \right) + 2\left( h - \tilde{h} \right) \sum_k w_k \delta(\mathbf{x} - \mathbf{x}_k)\, dV = 0
\tag{13}
$$

Applying integration by parts to the second term of the integrand and then applying (1) allows the simplification

$$
\int_\Omega \nabla \cdot (K \nabla \phi^*) + 2\left( h - \tilde{h} \right) \sum_k w_k \delta(\mathbf{x} - \mathbf{x}_k)\, dV = 0
\tag{14}
$$

## 2.4 Adjoint equation form

Inspired by (14), we propose to define $\phi^*$ to solve the BVP

$$
\nabla \cdot (K \nabla \phi^*) + 2 \sum_k w_k \left( h(\mathbf{x}_k) - \tilde{h}(\mathbf{x}_k) \right) = 0 \text{ in } \Omega,
\tag{15}
$$

$$
\phi^* = 0 \text{ on } \delta\Omega,
\tag{16}
$$

which we note is consistent with the adjoint BVP derived in Sykes et al. (1985). When $\phi^*$ is so defined, it eliminates two of the boundary integral terms and one of the volume integral terms in (12), leaving the simplified sensitivity equation

$$
\begin{aligned}
\frac{\partial J}{\partial \mathbf{p}} =& \int_\Omega 2\left( K \|\nabla h\| - \tilde{q} \right) \frac{\partial K}{\partial \mathbf{p}} \|\nabla h\| \sum_j w_j \delta(\mathbf{x} - \mathbf{x}_j)\, dV \\
&+ \int_\Omega \frac{2}{K} \frac{\partial K}{\partial \mathbf{p}} \left( \ln K - \ln \tilde{K} \right) \sum_l w_l \delta(\mathbf{x} - \mathbf{x}_l)\, dV + \frac{\partial R(K)}{\partial \mathbf{p}} \\
&- \int_\Omega \frac{\partial K}{\partial \mathbf{p}} \nabla h \cdot \nabla \phi^*\, dV - \int_{\delta\Omega} K \phi \nabla \phi^* \cdot dS.
\end{aligned}
\tag{17}
$$

We show here the specific dependence of the regularization term on the $K$ field alone.





## 2.5 Sensitivity to specific parameters

The vector $\mathbf{p}$ may contain two types of parameters: those defining the Type I (specified head) boundary condition, and those defining field $K(\mathbf{x})$, itself. We note that for any given parameter, $p_i$, the sensitivity expression (17) simplifies.

**For parameters, $p_i$, defining the conductivity field**, we observe that, because our domain has a specified-head boundary condition *which is governed by a different set of parameters* $\phi_i = 0$ in the final term of (17). Thus,

$$
\begin{aligned}
\frac{\partial J}{\partial p_i} = & \int_{\Omega} 2\left(K \|\nabla h\| - \tilde{q}\right) \frac{\partial K}{\partial p_i} \|\nabla h\| \sum_j w_j \delta(\mathbf{x} - \mathbf{x}_j) \, dV \\
& + \int_{\Omega} \frac{2}{K} \frac{\partial K}{\partial p_i} \left(\ln K - \ln \tilde{K}\right) \sum_l w_l \delta(\mathbf{x} - \mathbf{x}_l) \, dV + \frac{\partial R(K)}{\partial p_i} \\
& - \int_{\Omega} \frac{\partial K}{\partial p_i} \nabla h \cdot \nabla \phi^* \, dV,
\end{aligned}
\tag{18}
$$

There are at least two convenient ways that the conductivity field can be parameterized. The first way is that the domain can be discretized, with $p_i$ representing the constant value of $K$ in some subdomain, $\Omega_i$. In this case, $\frac{\partial K}{\partial p_i}$ is 1 in $\Omega_i$, and 0 outside. Thus,

$$
\begin{aligned}
\frac{\partial J}{\partial p_i} = & \int_{\Omega_i} 2\left(K \|\nabla h\| - \tilde{q}\right) \|\nabla h\| \sum_j w_j \delta(\mathbf{x} - \mathbf{x}_j) \, dV \\
& + \int_{\Omega_i} \frac{2}{K} \left(\ln K - \ln \tilde{K}\right) \sum_l w_l \delta(\mathbf{x} - \mathbf{x}_l) \, dV \\
& - \int_{\Omega_i} \nabla h \cdot \nabla \phi^* \, dV + \frac{\partial R(K)}{\partial p_i},
\end{aligned}
\tag{19}
$$


where only measurements inside the sub-domain participate in the sensitivity expression.

The second convenient approach is to write $\ln K(\mathbf{x}) = \sum_i p_i f_i(\mathbf{x})$, where the functions $f_i(\mathbf{x})$ form an orthogonal basis on $\Omega$. This Fourier series type of representation has the advantage of being independent of any domain discretization. Here, $\frac{\partial}{\partial p_i}\{\ln K\} = f_i(\mathbf{x})$, so we may adapt (18)

$$
\begin{aligned}
\frac{\partial J}{\partial p_i} = & 2 \sum_j w_j \left[ f_i \left(K \|\nabla h\| - \tilde{q}\right) K \|\nabla h\| \right]\bigg|_{\mathbf{x} = \mathbf{x}_j} \\
& + 2 \sum_l w_l \left[ f_i \left(\ln K - \ln \tilde{K}\right) \right]\bigg|_{\mathbf{x} = \mathbf{x}_l} \\
& - \int_{\Omega} K f_i \nabla h \cdot \nabla \phi^* \, dV + \frac{\partial R(K)}{\partial p_i}.
\end{aligned}
\tag{20}
$$


**For parameters defining the specified-head boundary condition**, it follows that $\frac{\partial K}{\partial p_i} = 0$, so all but the final term of (17) disappear. Invoking (8), it follows immediately that:

$$
\frac{\partial J}{\partial p_i} = -\int_{\delta\Omega} K \frac{\partial f}{\partial p_i} \nabla \phi^* \cdot dS.
\tag{21}
$$



As in the case of the conductivity field, the controlling parameters may either specify constant heads along individual boundary segments, or a series-type representation may be used. If $\Omega$ is convex, it is convenient to use a classic Fourier series, with an angular variable $\theta \in [0, 2\pi)$ sweeping the boundary like the hand of a clock.

## 3 Monte Carlo study of calibration

### 3.1 Overview

Using the theoretical derivations above, we study how well hydraulic conductivity and head fields can be reconstructed given point measurements of conductivity, head, and (optionally) groundwater velocity using adjoint-based gradient descent approaches. We aim to understand the relationship between the residual error of the fitted model and the $L^2$ (root integral square) error of the reconstructed conductivity field, and the degree to which this can be improved by the addition of (a) groundwater velocity information and (b) regularization based on statistical information about the conductivity field.

We approach this question via a Monte Carlo study: generating many synthetic target "true" conductivity realizations, and for each target realization, performing gradient descent model calibration multiple times from different random initial guesses. In this way, we are able to assess the relative performance of different calibration methods, as well as of measurement resolution.

We employ two specialized techniques: (i) series expansion in K-L basis functions, which addresses the problem of data generation as well as model-independent conductivity field representation, and (ii) adjoint-state inversion by steepest descent from initial guesses generated using the approach in (i). This addresses the issue of unbiased sampling of the model space as well as the computational tractability, as the computational cost of the adjoint-state approach is dictated by the dimension of the observation space, rather than the (much larger) dimension of the model space for under-determined inverse problems.

Any Monte Carlo study must necessarily be simplified relative to reality. While we have simplified the problem to make it more amenable to mathematical analysis, we expect the patterns that emerge to hold more broadly. Although our mathematical analysis is fully general for steady-state groundwater flow, we restrict the current study to reconstruction to two spatial dimensions and employ a separable exponential covariance structure for which the K-L basis functions may be analytically determined in closed form. Explicit expansion of the fields in these basis functions is advantageous, because it allows straightforward frequency-domain interpretation. We also assume that head boundary conditions are known, encoding a physically reasonable hydraulic gradient in the $x$-direction. We assume that a complete set of error-free, uniformly spaced measurements are available, allowing analysis to focus directly on the information content of the measurements and error induced by aliasing of unresolved high-frequency features. We further assume that true log-conductivity fields are Gaussian with a known correlation structure (equivalently, semivariogram), although except when regularization was employed, the statistics are only used for specification of initial guesses.





(a)

(b)

**Figure 1.** Examples of automated calibration (center column) of true fields $\ln \tilde{K}$ and $\tilde{h}$ (left column) after 500 optimization steps starting from an initially uncorrelated $\ln K$ field (right column), using the adjoint-state approach detailed in this paper. Color maps represent spatial variation of field values: higher value corresponds to lighter color. Black dots in each image indicate the locations of simulated measurements. Two calibrations are shown, both featuring the same underlying field statistics and the same data weights, as defined in (48-50, but differing in regularization approach: (a) regularization employed with $w_R = 1/2$, and (b) no regularization ($w_R = 0$). $\Psi = 0.10$ in both cases.



## 3.2 Representation of conductivity field and boundary conditions

We employ a generalized Fourier series type of representation of target log-conductivity fields as well as calibrated approxi-
mations to the target, meaning that fields are represented as vectors of coefficients and model calibration proceeds by iterative
solution of (15-16) and (20). The analytic representation of the adjoint-state gradient descent equations combined with the
series field representation means that any numerical discretization is completely abstracted out of the analysis, and calibration
can be performed by a general finite element solver, in our case the FEniCS package for Python.

The actual orthonormal basis functions used are the K-L basis functions corresponding to covariance kernel (22)

$$\text{Cov}(\ln K(\mathbf{x}_1), \ln K(\mathbf{x}_2)) = \sigma_{\ln K}^2 \exp\left(-\frac{|x_1 - x_2|}{\eta_x} - \frac{|y_1 - y_2|}{\eta_y}\right), \tag{22}$$

where the position vectors, $\mathbf{x}_i = \langle x_i, y_i \rangle$. Use of K-L basis function expansion has been recommended and previously employed
successfully in the context of hydraulic inversion by Wang et al. (2021). On our 2D domain $\Omega = [0, L_x] \times [0, L_y]$, the $n$-th such
function may be defined analytically (Zhang and Lu, 2004):

$$f_n(x, y) = \frac{1}{\sqrt{(\eta_x^2 w_{x,n}^2 + 1) L_x/2 + \eta_x}} [\eta_x w_{x,n} \cos(w_{x,n} x) + \sin(w_{x,n} x)] \times \tag{23}$$

$$\frac{1}{\sqrt{(\eta_y^2 w_{y,n}^2 + 1) L_y/2 + \eta_y}} [\eta_y w_{y,n} \cos(w_{y,n} y) + \sin(w_{y,n} y)], \tag{24}$$

where $w_{x,n}$ and $w_{y,n}$ are respectively the $n$-th positive solutions to

$$(\eta_x^2 w_{x,n}^2 - 1) \sin(w_{x,n} L_x) = 2\eta_x w_{x,n} \cos(w_{x,n} L_x), \tag{25}$$

$$(\eta_y^2 w_{y,n}^2 - 1) \sin(w_{y,n} L_y) = 2\eta_y w_{y,n} \cos(w_{y,n} L_y). \tag{26}$$

Viewing $w_{x,n} L$ as a single argument to both sine and cosine, it is clear that there will be infinitely many solutions, with one
occurring in each interval $(k\pi, (k+1/2)\pi)$, where sine and cosine must cross, regardless of vertical scaling (with one extra
solution occurring near 0). As $w_{x,n}$ grows, the solution point where $\eta_x w_{x,n} \sin(\cdot) \approx 2\cos(\cdot)$ must, on scale grounds, become
very close to the zeroes of the sine function, namely the whole number multiples of $\pi$. A similar argument applies for $w_{y,n}$.
Thus, it makes sense to understand the K-L basis functions as analogous to the basis functions of Fourier series, with angular
frequencies defined by $w_{x,n}$ and $w_{y,n}$. This makes it easy to interpret the several K-L basis function in terms of feature scale.

K-L basis functions, by definition, form an orthonormal basis so that

$$\int_0^{L_x} \int_0^{L_y} f_m(x, y) f_n(x, y) \, dy \, dx = \delta_{mn}, \tag{27}$$

and behave in such a way that any stochastic series

$$\ln \tilde{K} \equiv \sum_{n=1}^{\infty} \zeta_n \sqrt{\lambda_n} f_n(x, y) \tag{28}$$





where $\zeta_n \sim N(0,1)$, and eigenvalues $\lambda_n$ are defined

$$\lambda_n = \frac{4\eta_x\eta_y\sigma_{\ln K}^2}{\left(\eta_x^2 w_{x,n}^2 + 1\right)\left(\eta_y^2 w_{y,n}^2 + 1\right)} \tag{29}$$

represents an equally likely realization of a multivariate Gaussian field with covariance structure (22). The eigenvalues, $\lambda_n$, decrease with increasing $n$, representing the decreasing importance of each subspace to explaining the variance of the Gaussian field.

For a truncated K-L series representation with largest coefficient $N$, we can derive an expression for the portion of the variance, $\theta$, accounted for as follows. We define the total variance as the expected value of the integral of $\ln \tilde{K}$ over the entire domain. It follows from the definition of the pointwise variance, $\sigma_{\ln K}^2$, that

$$\mathrm{E}\left[\int_0^{L_x}\int_0^{L_y}\left(\ln\tilde{K}\right)^2 dy\,dx\right] = L_x L_y \sigma_{\ln K}^2 \tag{30}$$

Substituting in series representation (28), changing summation and integration order, and applying (27) yields

$$\int_0^{L_x}\int_0^{L_y}\left(\ln\tilde{K}\right)^2 dy\,dx = \int_0^{L_x}\int_0^{L_y}\left(\sum_{n=1}^\infty \zeta_n\sqrt{\lambda_n}f_n(x,y)\right)^2 dy\,dx \tag{31}$$

$$= \sum_{m=1}^\infty\sum_{n=1}^\infty \zeta_m\zeta_n\sqrt{\lambda_m}\sqrt{\lambda_n}\int_0^{L_x}\int_0^{L_y}f_m(x,y)f_n(x,y)\,dy\,dx \tag{32}$$

$$= \sum_{n=1}^\infty \zeta_n^2\lambda_n. \tag{33}$$

Taking the expectation and observing that $\zeta_n \sim \mathcal{N}(0,1)$ yields a series expression for the total variance,

$$\mathrm{E}\left[\int_0^{L_x}\int_0^{L_y}\left(\ln\tilde{K}\right)^2 dy\,dx\right] = \sum_{n=1}^\infty \lambda_n. \tag{34}$$

Thus, by truncating the RHS of (34) to $N$ terms and normalizing it via the RHS of (30), we arrive at the expression

$$\theta = \frac{1}{L_x L_y \sigma_{\ln K}^2}\sum_{n=1}^N \lambda_n. \tag{35}$$

Both target and initial guess fields represented as length-$N$ coefficient vectors, representing the coefficients $\zeta_n$ in (28). Because in our studies the only parameters being fit are the coefficients representing the log conductivity field, we can use the same K-L basis function expansion form to explicitly represent our calibrated approximations of the true $\ln \tilde{K}$:

$$\ln K(x) = \sum_n p_n\sqrt{\lambda_i}f_n(x). \tag{36}$$

While all solutions are expressed in terms of the K-L functions of corresponding to the true covariance structure, this in itself does not prejudice the solution, as these functions form a complete basis. One may wonder if the initial guesses being drawn





from the same distribution as the true field in some way biases them towards the "correct" field. However, we are interested in how cost function $J$ varies with $L^2$ reconstruction error during calibration, not the absolute magnitude of each, and the adjoint-state calibration equations (15-16) and (20) do not make use of the correlation information. This metric should not depend on the statistical structure of the initial guesses, as long as they are uncorrelated with their corresponding target field.

### 3.3 Bayesian selection of misfit and regularization weights

The units of speed, conductivity, and head are incompatible, so the sets of coefficients $\{w_j\}$, $\{w_j\}$, and $\{w_k\}$ implicitly have different units. It is necessary to select defensible values for these, and to select the regularization term $R(\mathbf{p})$. To select these optimally, we adopt a maximum *a posteriori* Bayesian standpoint, and imagine that the goal of our calibration is to determine the most probable parameter vector $\arg\max\limits_{\mathbf{p}} \Pr(\mathbf{p}|\mathbf{m})$, where $\mathbf{m}$ is the vector of all point measurements. It follows from Bayes' formula that

$$\Pr(\mathbf{p}|\mathbf{m}) \propto \Pr(\mathbf{m}|\mathbf{p})\Pr(\mathbf{p}) \tag{37}$$

The maximum probability vector, $\mathbf{p}^{\mathrm{max}}$, equivalently minimizes the negative logarithm of the probability

$$\mathbf{p}^{\mathrm{max}} = \arg\min\limits_{\mathbf{p}} -\ln\Pr(\mathbf{m}|K(\mathbf{p})) - \ln\Pr(\mathbf{p}). \tag{38}$$

The first term on the RHS, the *likelihood*, will give rise to the weights $w_j$, $w_j$, and $w_k$, and the second term, the *prior*, will give rise to the regularization term. Let us consider it first. By nature of the K-L expansion, all of the coefficients are independent and identically distributed according to $\mathcal{N}(0,1)$,

$$\Pr(\mathbf{p}) \propto \prod_n \exp\left(-\frac{p_n^2}{2}\right), \tag{39}$$

$$-\ln\Pr(\mathbf{p}) = \sum_n \frac{p_n^2}{2} + \mathrm{constant}, \tag{40}$$

where the constant is irrelevant from the perspective of minimization. We can adopt a similar approach for the likelihood term, defining the misfit-to measurement quantities

$$\Delta_j \equiv K(\mathbf{x}_j;\mathbf{p})\,\|\nabla h(\mathbf{x}_j;\mathbf{p})\| - \tilde{q}(\mathbf{x}_j) \tag{41}$$

$$\Delta_k \equiv h(\mathbf{x}_k;\mathbf{p}) - \tilde{h}(\mathbf{x}_k) \tag{42}$$

$$\Delta_l \equiv \ln K(\mathbf{x}_l;\mathbf{p}) - \ln\tilde{K}(\mathbf{x}_l) \tag{43}$$

By a symmetry argument, all of these quantities are zero mean before optimization and, in the absence of regularization, remain so throughout optimization. This is seen via the following argument: As of the initial guess, $\mathbf{p}^0$, they are computed by applying some fixed deterministic function to two independent, identically distributed random vectors ($\mathbf{p}^0$ and $\zeta$) and taking the difference between them. The expected value of this is zero. For any given pairing of true coefficient vector $\zeta$ and initial



guess vector $\mathbf{p}^0$, the pairing $-\zeta$ and $-\mathbf{p}^0$ is equally probable. Because the loss function only "sees" the squared deviation, it will respond in a symmetrical way to the original pairing as to the negated pairing: if `ParamIterStep`$(\zeta, \mathbf{p}^0) \to \mathbf{p}^1$, then `ParamIterStep`$(-\zeta, -\mathbf{p}^0) \to -\mathbf{p}^1$. Here, `ParamIterStep` represents the algorithm employed to iterate (improve) the parameter vector once by gradient descent. Thus, the expected value of the update vector is the zero vector. Inductively, the expected value of the deviations remains zero after any number of steps. Only the directional bias of a regularization term

breaks the symmetry.

The variances of these quantities will vary in an unpredictable over the course of model calibration. Before calibration, by symmetry, we expect, e.g., $\mathrm{Var}(\Delta_j) = 2\,\mathrm{Var}(\tilde{h})$. After successful calibration, $\mathrm{Var}(\Delta_j) \approx 0$. In general, we approximate $\mathrm{Var}(\Delta_j) \approx \sigma_{\tilde{q}}^2$, $\mathrm{Var}(\Delta_k) \approx \sigma_{\tilde{h}}^2$, and $\mathrm{Var}(\Delta_k) \approx \sigma_{\ln \tilde{K}}^2$.

Assuming $\Delta_j$, $\Delta_k$, and $\Delta_l$ are Gaussian distributed, we write

$$\Pr(\mathbf{m}|\mathbf{p}) \propto \prod_j \exp\left(-\frac{\Delta_j^2}{2\sigma_{\tilde{q}}^2}\right) \prod_k \exp\left(-\frac{\Delta_k^2}{2\sigma_{\tilde{h}}^2}\right) \prod_l \exp\left(-\frac{\Delta_l^2}{2\sigma_{\ln \tilde{K}}^2}\right). \tag{44}$$

$$-\ln \Pr(\mathbf{m}|\mathbf{p}) = \sum_j \frac{\Delta_j^2}{2\sigma_{\tilde{q}}^2} + \sum_k \frac{\Delta_k^2}{2\sigma_{\tilde{h}}^2} + \sum_j \frac{\Delta_j^2}{2\sigma_{\ln \tilde{K}}^2} + \text{constant}. \tag{45}$$

Combining (37), (40), and (45) and inserting the definitions for $\Delta_j$, $\Delta_k$, and $\Delta_l$ gives us the expression to minimize

$$
\begin{aligned}
-\ln \Pr(\mathbf{m}|\mathbf{p}) = &\sum_j \frac{1}{2\sigma_{\tilde{q}}^2}\left(K(\mathbf{x}_j;\mathbf{p})\|\nabla h(\mathbf{x}_j;\mathbf{p})\| - \tilde{q}(\mathbf{x}_j)\right)^2 \\
&+ \sum_k \frac{1}{2\sigma_{\tilde{h}}^2}\left(h(\mathbf{x}_k;\mathbf{p}) - \tilde{h}(\mathbf{x}_k)\right)^2 \\
&+ \sum_j \frac{1}{2\sigma_{\ln \tilde{K}}^2}\left(\ln K(\mathbf{x}_l;\mathbf{p}) - \ln \tilde{K}(\mathbf{x}_l)\right)^2 \\
&+ \sum_n \frac{1}{2}p_n^2 + \text{constant}.
\end{aligned}
\tag{46}
$$

The constant is irrelevant from the perspective of minimization. By comparison with loss function $J$ defined in (3), we are thus motivated to define

$$R(\mathbf{p}) \equiv w_R \sum_n p_n^2, \tag{47}$$

and to define the various weighting coefficients.

$$w_j = \frac{1}{2\sigma_{\tilde{q}}^2} \qquad\qquad \forall j, \tag{48}$$

$$w_k = \frac{1}{2\sigma_{\tilde{h}}^2} \qquad\qquad \forall k, \tag{49}$$

$$w_l = \frac{1}{2\sigma_{\ln \tilde{K}}^2} \qquad\qquad \forall l, \tag{50}$$

$$w_R = \frac{1}{2}. \tag{51}$$





The relevant variances were determined by generating an ensemble of 20 realizations via K-L expansion, solving the ground-water flow equation (1-2) on them, and then generating simulated measurements. At each location, for each of the three types

of measurement, the average value was computed, and the ensemble variances, $\sigma^2_{\ln K}$, $\sigma^2_{K||\nabla h||}$, and $\sigma^2_h$ were computed for each value about its appropriate mean value.

### 3.4 Mechanics of Monte Carlo study

A standard rectangular field was employed for all simulations, with dimensions $L_x = 10$, $L_y = 5$, and all simulated fields featured the same separable 2D correlation structure as described in Section 3.2, with $\eta_x = \eta_y = 2$ and $\sigma^2_{\ln K} = 2$, by K-L

expansion. Enough basis functions were employed to account for 98.7% of the variance: the 2000 K-L basis functions with the largest eigenvalues were employed to generate the random fields. Any given realization is completely represented by a 2000-component vector, $\zeta$ containing its K-L coefficients. The length scales $L_x$ and $L_y$ are artifacts of representing a conceptually infinite random field on a computer, and we aim to select it large enough that its value does not affect our conclusions about the information content of a given density of point measurements. Apart from $L_x$ and $L_y$, the only physically significant length

scales are the correlation lengths, $\eta_x$ and $\eta_y$, and these are used in our analysis.

For each realization, simulated measurements were computed on uniformly-spaced square lattices with point separation $\Delta_m$ by solution of (1-2) using the FEniCS finite element system (Logg et al., 2012), employing a 100 by 50 finite element grid with first-degree accurate square finite elements. Measurement density relative to field correlation length is represented by the dimensionless quantity

$$\Psi \equiv \frac{\Delta^2_m}{\eta_x \eta_y}. \tag{52}$$

The main MC study explored field reconstruction for ensembles of fields corresponding five different values of $\Psi$, covering measurement spacings, $\Delta_m$, ranging from dense to sparse relative to field correlation length. This study used a full complement of measurements, with values of speed, conductivity, and head at each measurement location, and also featured regularization. A second MC study was performed, using only the intermediate value $\Psi = 0.1$, which considered the impact of removing

some information, with three ensembles neglecting one of regularization ($w_R = 0$), speed ($w_j = 0$), and conductivity ($w_l = 0$), respectively. A straightforward regularization-by-truncation scheme in which only leading (large eigenvalue) terms were calibrated and $w_R = 0$ was also considered.

Ensemble size varied somewhat between ensembles, but at minimum each ensemble contained at least 50 unique "true" random fields from which simulated measurements were made, and calibration was run for each from at least two random

initial parameter sets, resulting in at least 100 automated calibrations. Each calibration was performed by iterative evaluation of (15-16) and (20), over 500 steps (counted by runs of the forward model), using a straightforward adaptive step size algorithm which was found to work well: grow the step size by 10 % after each step down gradient that successfully reduces the loss function, (3), and otherwise halve it until descent along the gradient reduces the loss function. Each time the algorithm achieves a new minimum of the loss function, its value is stored, along with the parameter vector, $\mathbf{p}^i$, where $i$ is the parameter iteration

index. Iterations were counted only when there was a reduction in the loss function (once for each run of the adjoint model),





so typically slightly fewer than 500 parameter iterations were stored due to occasional step size cuts that required additional forward model runs. The initial random parameter guess vector $\mathbf{p}^0$ is also stored for each calibration. The parameter vector corresponding to the smallest loss function is given the symbol $\mathbf{p}^{\text{final}}$, and used alongside $\mathbf{p}^0$ to quantify the improvement in the coefficient of each K-L basis function as a result of the automatic calibration. For performance reasons, fewer K-L

functions were used in calibration than were used to generate the measurements, with the coefficients for the least significant (smallest eigenvalue) K-L functions fixed at zero. We denote the number of coefficients actually calibrated by $\nu$. Typically, $\nu = 1350$ coefficients, representing 98.2% of variance, were employed, except in one trial, in which only $\nu = 12$ coefficients, representing 62.7% of variance were calibrated. The motivation for this selection depends on results from the main MC study, and is discussed in Section 4.

**3.5   Scale and fitness metrics**

To quantify the results of the parametric study, it is helpful to define some quantities relating to feature scale and to the accuracy of the reconstruction.

The eigenvalues, $\lambda_n$, are proxies for the significance of the various basis functions in the K-L expansion, but are difficult to interpret physically. For analysis, we define *normalized basis function period* as the harmonic mean of the $x$- and $y$-direction

periods, normalized to the measurement spacing, $\Delta_m$:

$$T_n = \frac{1}{\Delta_m} \frac{2}{\frac{1}{2\pi w_{x,n}} + \frac{1}{2\pi w_{y,n}}}. \tag{53}$$

This quantity is a measure of the feature scale associated with K-L basis function $\phi_n(x,y)$, and was found to vary nearly monotonically, though non-linearly with eigenvalue. Precisely, if $T_2 > T_1$, then $\lambda_2 + \epsilon > \lambda_1$, for some small $\epsilon$.

To quantify the reconstruction error, we will generally employ the $L^2$ (root-mean square) norm, $||\cdot||_2$, to quantify the distance

between the true log-conductivity field, $\ln \tilde{K}$, and the calibrated log-conductivity field, $\ln K$:

$$\left\| \ln \tilde{K} - \ln K \right\|_2 \equiv \sqrt{\int_0^{L_x} \int_0^{L_y} \left( \ln \tilde{K}(x,y) - \ln K(x,y) \right)^2 dy \, dx} \tag{54}$$

$$= \sqrt{\sum_n \lambda_i (\zeta_n - p_n)^2}, \tag{55}$$

where the last equality follows from Parseval's theorem. Implications of the theorem are that we can work directly with the coefficient vectors $\zeta$ and $\mathbf{p}$, and that we can thereby quantify the proportion of the error in the subspaces defined by a given

K-L basis function or collection of basis functions. As we are especially interested in the reconstruction of features with large scales, which are captured by basis functions with large $T$ (equivalently large eigenvalues, which have low indexes, $n$), we define

$$L_N^2 \equiv \sqrt{\sum_{n=1}^{N} \lambda_i (\zeta_n - p_n^{\text{final}})^2}, \tag{56}$$



where $p_n^{\text{final}}$ is the $n$-th component of $\mathbf{p}^{\text{final}}$. Where all components are employed ($N = 2000$), we sometimes drop the subscript, $N$, for clarity. We also define a normalized form of this quantity

$$\hat{L}_N^2 \equiv \frac{L_N^2}{\frac{\sum_{n=1}^N \lambda_n}{\sqrt{L_x L_y \sigma_{\ln K}^2}}}, \tag{57}$$

which has the interpretation of root-mean-square error divided by the proportion of the $L^2$ norm of the entire log-conductivity field, i.e., $\|\ln K\|_2$ included in the considered subspace. We are also interested in the amount of learning that occurs in the subspace defined by each K-L basis function over the course of calibration. We define

$$r_n \equiv \left\langle \frac{(\zeta_n - p_n^{\text{final}})^2}{(\zeta_{n,0} - p_n^0)^2} \right\rangle_G, \tag{58}$$

where $p_n^0$ is the $n$-th component of initial coefficient vector $\mathbf{p}^0$, and $\langle \cdot \rangle_G$ represents the geometric mean over all the calibration trials in an ensemble. If $r_n = 1$, measurements on average provided no information about the coefficient for K-L basis function $f_n(x,y)$, and if $r_n = 0$, then perfect identification occurs.

## 4 Results and discussion

### 4.1 Reduction in global $L^2$ error by calibration

Because the loss function is based on point and / or proxy measurements for the true fields of interest, the relationship between reductions in the loss and improved accuracy of the underlying $\ln \tilde{K}$ field is only indirect. Head fields are generally smooth relative to their underlying conductivity fields, observations are sparse, and conductivities along a streamline may be permuted without altering the effective conductivity along that streamline: permutation is detectable only indirectly through the deformation of neighboring streamlines. Thus, it is not *a priori* obvious how well calibration of point measurements (i.e., reducing loss function $J$) will succeed in constraining conductivity values remote from the measurement locations. To examine this, we consider the global $L^2$ error of the entire reconstructed conductivity field, $\ln K$, relative to the known field, $\ln \tilde{K}$ from which the measurements are generated. This metric weights accuracy at all locations in the domain equally, so if reduction in the loss function truly results in learning about the underlying field, reduction in $J$ should correspond to reduction in $L^2$. In Figure 2, we present results comparing the global error, $L^2$ with $J$ for each iteration in each automated calibration for each of five ensembles in the main MC study featuring different measurement densities from $\Psi = 0.02$ to $1.56$. In general, a steep decline was observed in $L^2$ versus $J$ for early iterations, which plateaued in later iterations, as further reduction in the objective function ceased to cause significant improvements in model fidelity. The ultimate level of $L^2$ error at the plateau was found to increase with $\Psi$, as the average distance to the nearest measurement point increased and the measurement set contained less relevant information to constrain the conductivity and head fields there.





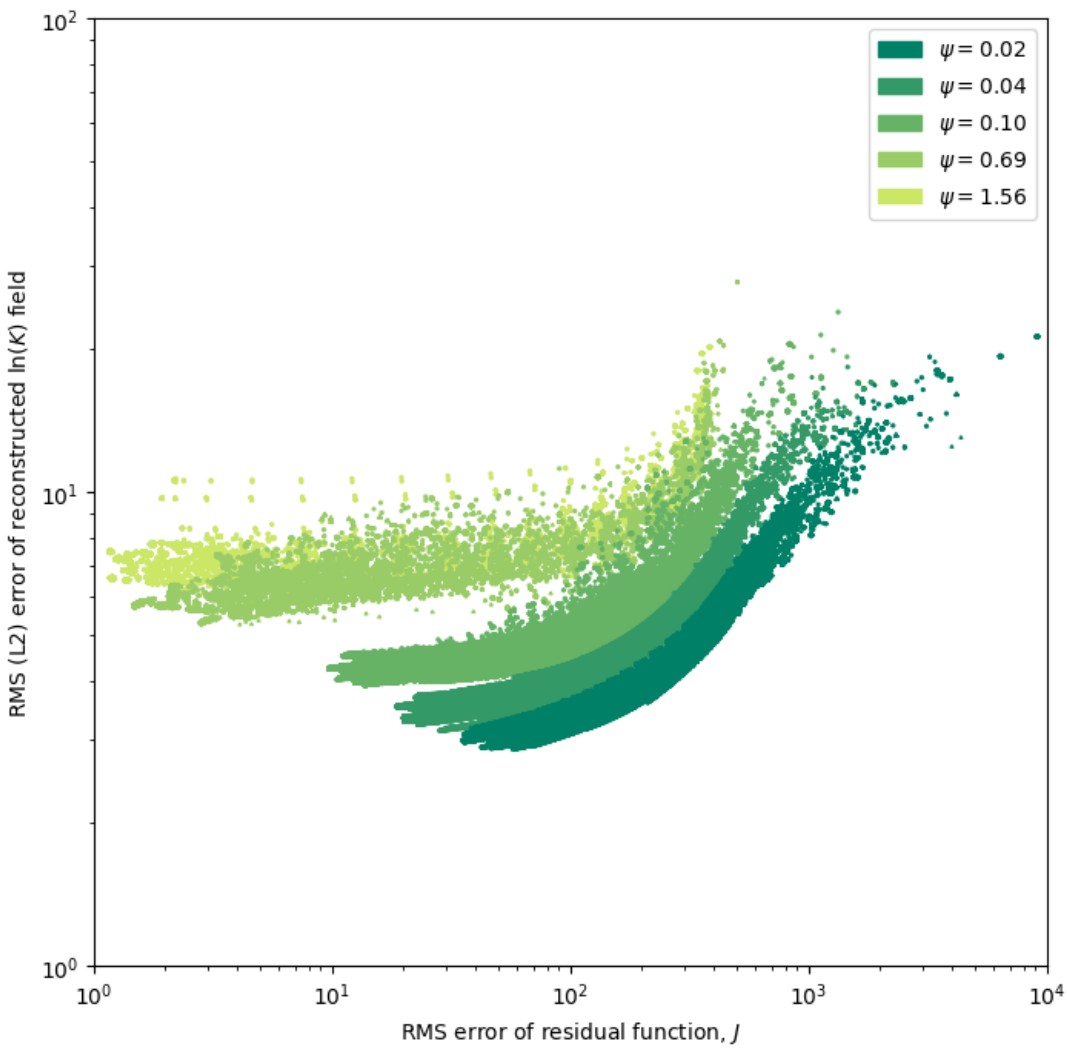

**Figure 2.** Comparison of the $L^2$ error of the reconstructed hydraulic conductivity field versus the residual function, $J$, defined in (3) for several different measurement densities, $\Psi$, in the main MC study. Weights are as defined in (48-51). Each ensemble is plotted with a unique color, and within each ensemble, each parameter iteration for each calibration trial (i.e., unique pairing of $\zeta$ and $\mathbf{p}^0$) is shown.





## 4.2 Feature scale and information gain due to calibration

A second illuminating analysis that can be performed on the main MC study is to consider the average improvement each individual K-L coefficient, indexed by $T$, the basis function period normalized to the measurement spacing. This allows us to determine the ability to identify features of a particular scale by calibration, and to determine a cutoff beneath which small-scale variability is essentially invisible to the calibration process. In 3, the average ratio of squared coefficient error after vs. before calibration is shown at a function of $T$ for each calibrated K-L basis function, for all five ensembles, each for a different value of $\Psi$. for small $T$, we observe a relatively flat (relative to $T$) regime in which only minor improvement is observed. We attribute the improvement to the $w_R$ regularization term, for reasons discussed below. Around $T = 4$, we observe a sharp regime change, in which $r$ begins to rapidly decrease in power law fashion with increasing $T$, we identify the following empirical relation for this regime,

$$r_n \approx 10\, T_n^{-2.5}, \qquad T_n > 4,$$ (59)

shown as the dotted line in Figure 3. A consequence of this analysis is that positive identification becomes possible only once feature scale exceeds four times the measurement spacing. We note good coherence between the various ensembles, suggesting that $T$ is an adequate metric for capturing relative basis function scale.

## 4.3 Information gain due to velocity measurements and regularization

In the second MC study, we fixed $\Psi = 0.10$ but modified the objective function in various ways to ascertain how the modifications affect the $L^2$ accuracy of the ultimate reconstruction ($L_{2000}^2$), and also the ability to calibrate large scale features in the 12-dimensional subspace spanned the K-L basis functions satisfying the previously identified criterion $T > 4$ ($L_{12}^2$). The specifics of the studies and the resulting metrics are tabulated in Table 1. A visual representation of the amount of learning that occurs about the coefficients of each K-L basis function, sorted by $T$, analogous to the one for the main study is shown in Figure 4.

In the base case (featuring all available information) was shared with the main MC study. In this data set, calibration was performed beginning from a pair of different random $\mathbf{p}^0$ vectors for each true field vector, $\zeta$. Visual examination of the calibrated head fields for this data set showed that the fitted $\ln K$ and $h$ fields were in all cases visually identical for both of the paired initial parameter guesses. This suggests that regularized objective function was convex, and that the unique solution is essentially found in all cases, with no influence of initial conditions. However, even when the unique regularized solution is achieved, the remaining relative error, $\hat{L}_{2000}^2$, is still $42\%$, and $12\%$ in the subspace spanned by the most significant (largest period) basis functions (for more details) . Figure 1 (a) visually illustrates the results: note how large-scale features are captured, but there is still significant infidelity between the smooth, regularized $\ln K$ and the true $\ln \tilde{K}$.

Where regularization is disabled, pairwise visual comparison reveals some residual effect of initial guess vector, although large-scale features of the true fields are plainly always captured. Unregularized calibrated $L^2$ error was only slightly worse than for regularized calibration in the most significant subspace (again see Table 1), attributable to the fact that the data is highly




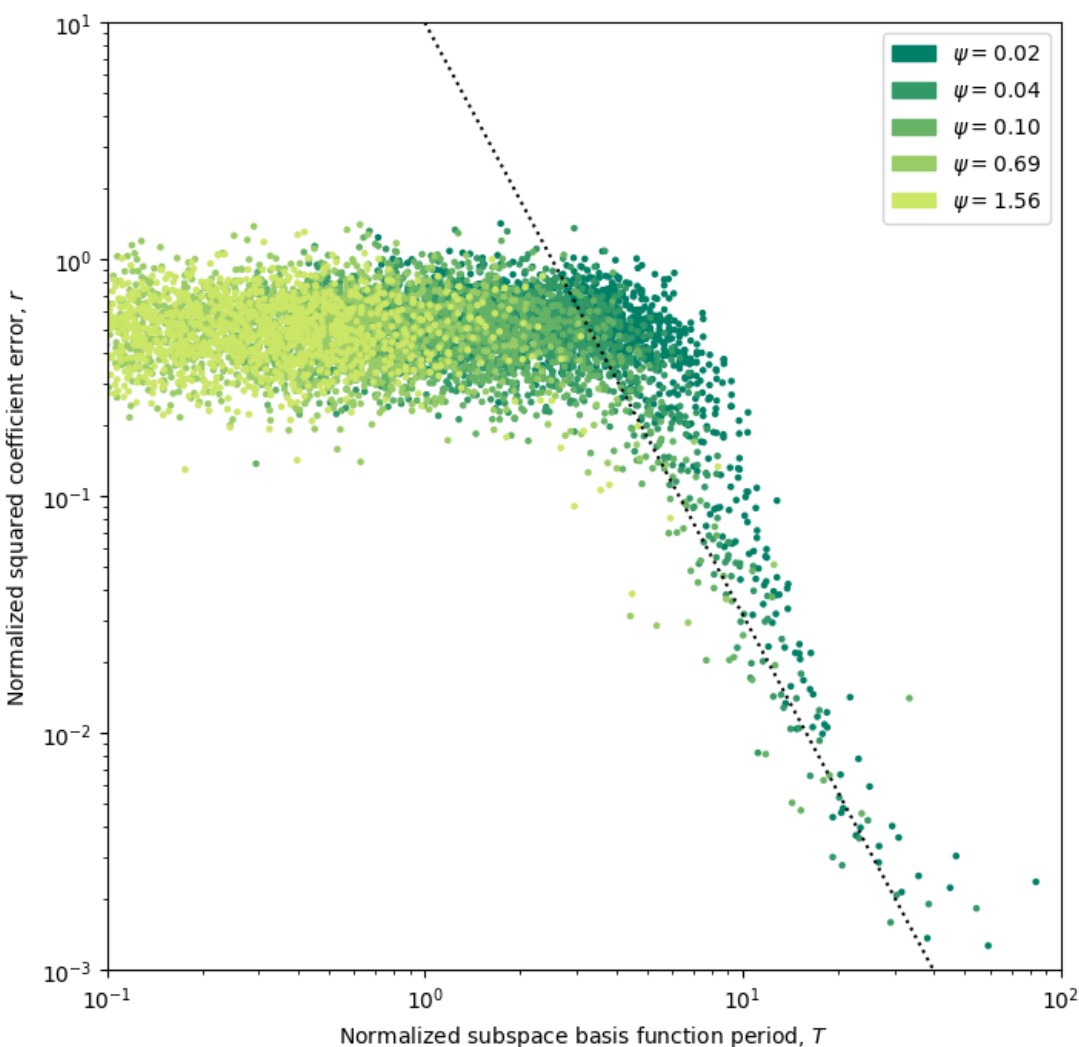

**Figure 3.** Geometric mean squared relative coefficient error improvement, $r$ for each K-L basis function, $f_n(x)$, across all calibration trials (i.e., unique pairing of $\zeta$ and $\mathbf{p}^0$) in each ensemble (corresponding to a different value of $\psi$) in the main MC study. Each basis function is represented by its normalized period, $T$, rather than its index. A dashed trend line scaling as $T^{-2.5}$ is shown in the large period region.



informative about these K-L coefficients, and there is limited work for regularization to accomplish. For the unregularized ensemble, we attribute residual $L^2$ error to errors in high-frequency subspaces from the initial guess $\mathbf{p}^0$ that manifest at

locations remote from the measurements and which are left over after large-scale features have been identified. This is an entirely different cause of error than the smoothing which generates unique solutions in the $w_R$-regularized case. Figure 1 (b) illustrates unregularized calibration.

Without regularization from $w_R$, $r \approx 1$ in the small-scale, $T < 4$ subspace, indicating that the measurements provide no information about these coefficients. We attribute the improved performance with the regularization term to the fact that the

coefficients of $\zeta$ and $\mathbf{p}^0$ are all iid random variables $\sim \mathcal{N}(0,1)$ it is easy to show that $\mathrm{Var}\,(\zeta_n - p_n^0) > \mathrm{Var}\,(\zeta_n) = 1$, so by pulling each $p_n$ towards zero, the $w_R$ term in $J$ decreases average square error of $p_n$, even absent any information from the measurements regarding its value.

A previous study (Tso et al., 2016), performed using a different co-kriging-based calibration approach suggested a significant gain in reconstruction reliability over use of head-conductivity co-kriging. We tested whether groundwater speed would provide

similar extra information in the context of gradient descent model calibration by performing a calibration ensemble in which its weight was zero. We found performance essentially identical to that for the base case. Given that head and conductivity fields jointly determine the flux everywhere, and regularization is sufficient to determine a unique solution, the fact that speed information was found to be redundant implies that the regularized solution correctly identifies $\tilde{h}$ and $\tilde{K}$ at the measurement locations.

Given the idea that a correct speed and head field could uniquely determine $\tilde{K}$ everywhere, we examined the extent to which speed measurements could substitute for hard-to-determine point measurements of $\tilde{K}$ by eliminating the weight on these measurements in $J$. This was found to be possible to some extent, but performance was significantly reduced relative to all other approaches, with identification in the most significant, large period subspaces, as measured by $L_{12}^2$, significantly degraded. A possible explanation for this result is that calibration of $\tilde{K}$ becomes sensitive to errors in $\nabla h$ as well as in $h$.

Finally, motivated by the idea the large-period subspace ($T > 4$) is readily identifiable without regularization, and that the measurements contain little or no information about the small-$T$ (small scale) subspace, we attempted a naive truncation-based regularization approach in which only the most significant, $T > 4$ K-L coefficients are calibrated, and the rest are fixed to 0, and no regularization term is applied (i.e., $\nu = 12$ and $w_R = 0$ for this ensemble). Performance of this approach was not as strong as the other trials, with $\hat{L}_{12}^2 = 0.21$, about 50% greater than in the base case, but identification of the large-scale subspace

remained respectable and still exceeded the regularized trial with no $\ln \tilde{K}$ measurements. When the K-L field representation is employed, $\partial K / \partial p_n$ does not collapse to unity, so there remains an expensive basis-function-specific integration in (20). Thus, the computational cost of computing the gradient using naive truncation was reduced more than 100-fold in this ensemble relative to the base case and this approach merits further consideration.




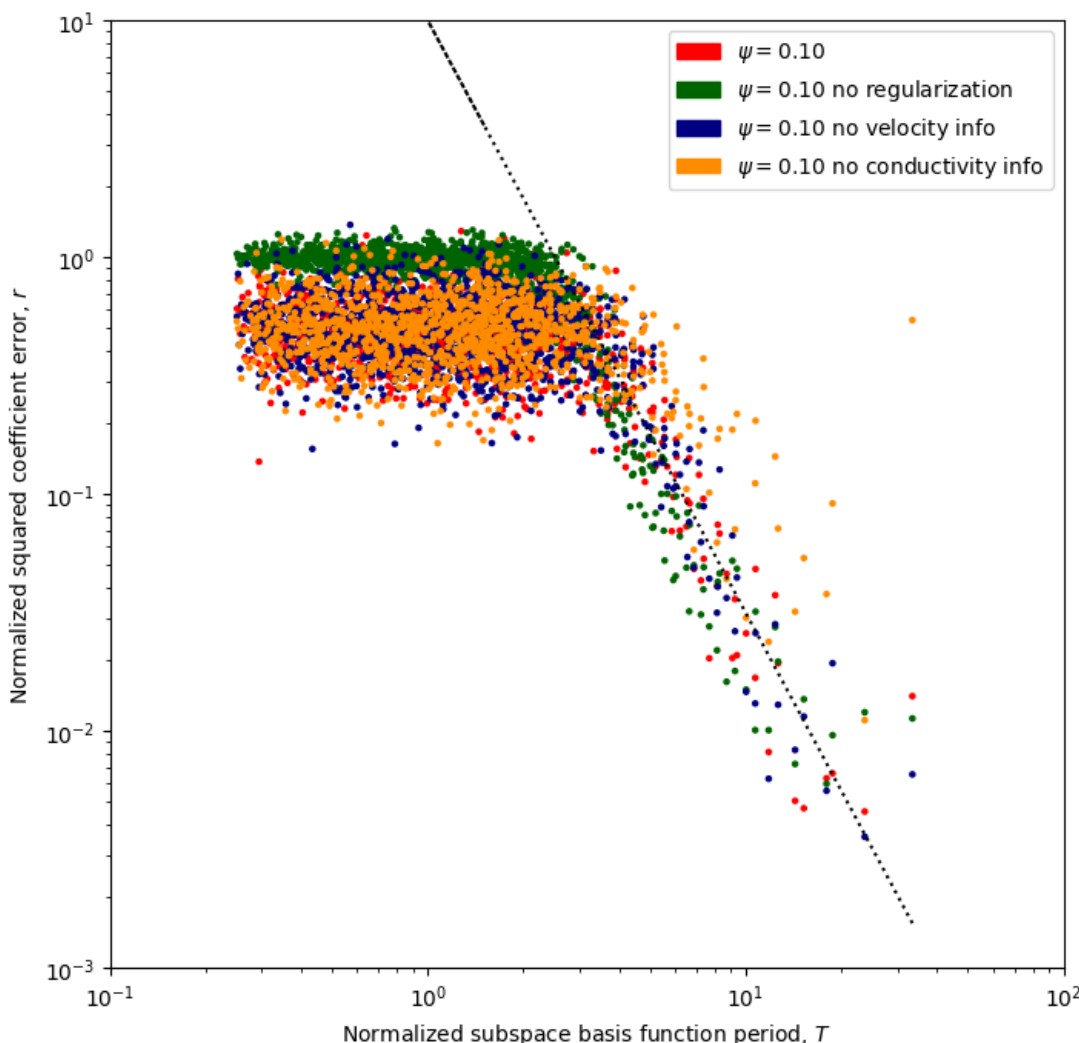

**Figure 4.** Geometric mean squared relative coefficient error improvement, $r$, across all calibration trials in each ensemble in the second MC study. Results are presented as a function of normalized K-L basis function period, $T$, with each ensemble representing different terms present in the loss function. A dashed trend line scaling as $T^{-2.5}$ is shown in the large period region.



**Table 1.** Descriptive statistics of five ensemble calibration experiments in the second MC study, corresponding to $\Psi = 0.1$

|  | $w_j$ | $w_k$ | $w_l$ | $w_r$ | $\nu$ | $L_{12}^2$ | $\hat{L}_{12}^2$ | $L_{2000}^2$ | $\hat{L}_{2000}^2$ |
|---|---|---|---|---|---|---|---|---|---|
| Base case | 2 | 225 | 0.25 | 0.25 | 1350 | 1.09 | 0.14 | 4.25 | 0.42 |
| No regularization | 2 | 225 | 0.25 | 0.0 | 1350 | 1.19 | 0.15 | 4.93 | 0.49 |
| No $\tilde{q}$ information | 0 | 225 | 0.25 | 0.25 | 1350 | 1.14 | 0.14 | 4.29 | 0.43 |
| No $\ln \tilde{K}$ information | 2 | 225 | 0 | 0.25 | 1350 | 3.58 | 0.45 | 6.04 | 0.60 |
| Truncated | 2 | 225 | 0.25 | 0.0 | 12 | 1.68 | 0.21 | 6.19 | 0.62 |

## 5   Summary and conclusions

We computed the sensitivity of a loss function based on three types of point hydraulic measurement and a general regularization term to an underlying, heterogeneous hydraulic conductivity field via a continuous adjoint-state approach. Using K-L expansion, we created a general, zonation-free technique for analyzing the quality of inversion from point data to an underlying heterogeneous scalar field. Using K-L basis functions which feature clearly-defined periods, we analyzed how reliably conductivity and head fields were recovered, both as a function of feature scale and globally, given different densities and types

of point measurements.

The following are our key conclusions:

➢ Substantially better than random identification accuracy is observed as feature scales exceed four times the measurement spacing, with relative squared error after calibration scaling as $T^{-2.5}$, where $T$ is the feature scale normalized to measurement spacing. This relation provides a useful guideline for needed measurement density to achieve desired field

accuracy. For higher resolution characterization, non-point geophysical or tracer data may be required.

➢ Even at very coarse measurement resolutions relative to $K$ field correlation length, the point-wise loss function is found informative globally (in an $L^2$ sense) about the underlying conductivity field, and calibration offers better than random feature identification at all scales when regularization is employed.

➢ Groundwater speed point measurements can partially substitute for point conductivity measurements, but contra the

sequential linear estimation study of Tso et al. (2016), provide little additional information in the context of gradient descent optimization with the adjoint-state approach, provided reliable head and conductivity point measurements already exist.

➢ Gradient descent calibration ultimate $L^2$ error is dependent on measurement density and displays quickly diminishing returns on optimization. We posit that large-scale features are identified quickly and the loss function is less sensitive to

point mismatch remote from the measurement locations. Thus, later iterations tend to optimize field values near the well: $K$-field infidelity remote from the measurement locations that exists by the time large-scale features have been identified tends to persist.



The approach adopted here has a number of useful features that may be helpful for further investigations. The analytical sensitivity expression for a loss function containing many different types of data and regularization can be used for model fitting, and its zonated form (19) is particularly efficient for use with finite volume and finite element codes. Also, the sort of zonation-free K-L series Monte Carlo calibration study could be used to explore the identifiability of other geophysical fields.

*Code and data availability.* Code and data for this project are available in a Zenodo repository, accessible at https://doi.org/10.5281/zenodo.10513647. Archived elements include the Python scripts necessary to generate the simulation outputs used in the analyses, the simulation outputs themselves, and the Python scripts used to generate the figures.

*Author contributions.* SKH and JPH performed conceptualization and funding acquisition. All authors contributed to research methodology and software. SKH and DM contributed to the formal analysis. SKH led the investigation and performed original draft preparation. DM contributed review and editing.

*Competing interests.* The authors declare that they have no conflict of interest.

*Acknowledgements.* This work was partially supported by seed funding from the Northwestern Center for Water Research and the Zuckerberg Institute for Water Research. SKH was supported by Israel Science Foundation personal research grant 1872/19. SKH holds the Helen Ungar Career Development Chair in Desert Hydrogeology.



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
