# Peer review of "Feature scale and identifiability: How much information do point hydraulic measurements provide about heterogeneous head and conductivity fields?"

_EGUsphere, 2024_

## Author Response (AR1)

**Response to comments on *Feature scale and identifiability: How much information do point hydraulic measurements provide about heterogeneous head and conductivity fields?**

In this document we reproduce the editor, reviewer, and community comments in  gray boxes , with our responses following, explaining our reasoning. Explanation of how specifically we have altered the MS in response to the comments appears in **bold**.

**Editor's remarks**

> Two referees provided their assessment: one was very positive, the second one stressed some weaknesses. Basically, the questions posed by the more critical referee concern the fact that the results and discussions cannot be generalized in a straightforward way and the conclusions could somehow depend on the specific hydrogeological setup considered for the numerical test case.
>
> I am well aware that any synthetic aquifer and hydrogeological setup used to test a new method can be subject to this kind of criticism. But, nevertheless, the authors should try to review sections 3 and 4, and conclusions in order to focus better which results can be considered of general validity and which results are more strictly dependent on the hydrogeological set-up considered for the test case. in fact, some sentences (e.g., lines 199 to 203) and the titles of the subsections in section 4 seem to introduce results of general validity.

We appreciate being given the opportunity to address these concerns.

We agree that the generality of the results is an important question which required further discussion. We do believe our results to be applicable beyond the specific system we have analyzed, however **in the revised manuscript, we have made some of our claims more precise, both in the lines identified, and in the conclusion. We now highlight more than once that our study is focused on a paradigmatic 2D steady-state flow scenario**. In this context, we believe it is clear that the headings in section 4 refer to the specific scenario we study. **We have also added a new subsection, 4.4 which addresses our understanding of the generality of our results and their applicability to other flow regimes**. We believe that some of the more critical reviewer's concerns may be related to their understanding of our paper as a model benchmark study rather than the basic science investigation that we understand it to be. Accordingly, **we have added material to the introduction that we believe better clarifies the nature and scope of the study** which may address some reviewer concerns.

We hope that our discussion in this document, as well as the new material added the manuscript will satisfy all outstanding concerns.

**Reviewer 1**

> The authors present an approach for using Karhunen-Loeve explanations to estimate spatial variability of properties and system states on observations of head, permeability, and velocity magnitude, both with and without regularization. They apply the adjoint approach for efficiently determining gradients of the objective function, allowing for more efficiency and, presumably, high-order approximations. The methods are explained thoroughly, including the assumptions and simplifications. I just have a few minor comments.

We appreciate the reviewer's overall positive assessment.

> 1. A substantial amount (about 20%) of the paper was devoted to developing the adjoint equation and the adjoint-based form of the derivatives of the objective function. Since that was just a tool to be used in the analysis, the detailed development seemed to detract from the main focus of the paper. Could Section 2 be moved to an appendix?

We agree that the flow of the paper is improved by moving many of the derivation details into an appendix. **We have moved the details of the derivation into the appendix**, retaining only the equations required for the main discussion in the body of the manuscript.

> 2. Figure 1 is very helpful as an example of the recovery of spatial distributions of head and permeability through the approach presented in the paper. All other figures show "error" between measures and fits, so Figure 1 is very useful as means of showing the reader the intermediate step. However, Figure 1 is out of place – it appears on p. 11, but it isn't mentioned in the text until page 20. Also more explanation can be provided to make the link between what appears in Figure 1 and how that is related to the data point plotted in the other figures. I wonder also why the two subplots of Figure 1 use different true ln K fields. If subplot a has regularization and subplot b does not, it would be more informative to see that results from the same true ln K field so that the reader can see the benefit of regularization. It would also be helpful to see the numerical value of "error" (the quantities that are plotted in the other figures) to get a sense for where these fits fall in those plots, compared to all other realizations that appear in the plots in Figures 2 and above.

**We have added reference to Figure 1 in the introduction in our revised explanation of the aims and scope of the analysis. We have also revised Figure 1 so that both the fitting examples employ the same ground truth. We now also report and discuss the L2 error fitting error associated with the calibrations.**

> 3. I would like to see some explanation of the practicality of this method. What measurement density is needed? Is it different for measurements of different quantities?

In our view, one of the most interesting outcomes of this work was the result that the features become identifiable as measurements become denser than 1/4 the feature scale. We also consider some different data (and regularization) combinations in Table 1, Figure 4, and the surrounding discussion. We think that these results may have been under-emphasized, so **we have augmented the discussion and added material to the introduction emphasizing these results**.

**Reviewer 2**

This is a well written paper (until the discussion section) on the study of the worth of different types of data for inverse modeling in hydrogeology. Its aim is to try to identify the necessary sampling density in order to identify the underlying hydraulic conductivity structure. While I enjoyed reading the paper and the detailed derivations of the different equations, I had a hard time trying to find the novelty of the work. This question has been studied as early as in the late 1990s,

We are very surprised by these remarks. To our knowledge, our paper has two major original features:

1. The principal novelty of our study is the investigation of the connection between spatial scale of features in the hydraulic conductivity field and our ability to characterize them from a given density of point measurements. As far as we know, this has never been systematically investigated and no one has previously quantified the relationship between amount learned and relative measurement density, as we have. Our key result—that feature identification becomes possible above 4x measurement scale, with quality increasing in power law fashion with increasing feature scale above this threshold—has no real precedents.
2. The Karhunen-Loeve basis function framework we develop for assessing calibration accuracy without requiring the imposition of experimenter-specified a priori spatial zonation or subjective weights on the objective function is itself novel and can be applied (with suitable modifications) to analysis of a variety of different field recovery problems.

The reviewer did not respond to a request for elaboration as to what work in the 1990s they believe anticipated our analysis. Again, no previous study we know of has systematically considered the relationship between feature scale and feature identifiability by means of model calibration.

In reaction to these comments, as well as to some remarks by the first reviewer, **we have augmented our introduction section to clarify our science goals and key results, and to emphasize what is novel**.

and the application proposed, while very elegant, is limited to a very narrow and little interesting case of steady groundwater flow in an aquifer with prescribed head at the boundary and with an underlying hydraulic conductivity field drawn from a multi Gaussian random function with an exponential covariance. For the paper to be worth to be considered for publication it should have addressed a tougher problem: transient, non-Gaussian field, larger variance, anisotropic, generic boundary conditions, addressing the issue of measurements taken at different supports. As is, the a paper is a very nice mathematical exercise with little practical interest.

We are surprosed to see the argument that the manuscript would only be worthwhile to publish if it considered a "tougher problem" with, e.g., an esoteric non-Gaussian statistical correlation structure, or high variance, or transient flow. This seems strange and perhaps represents a misunderstanding of what we are studying.

If our work were an algorithm benchmarking study where we were trying to select the most robust approach from amongst candidates, we could see the merit in choosing a "tough" benchmark. But our work is not a benchmarking study; rather, it concerns the information content of hydraulic measurements. This basic science question is best answered using a canonical flow model that exhibits the physics we wish to study, employing realistic

parameters and standard assumptions but otherwise shedding needless complexity. Therefore, the more niche modeling scenarios suggested by the reviewer are counterproductive for our purposes. We stress: our study is not "about" any specific flow configuration; rather we are studying how reliably groundwater flow transports information about conductivity features to remote measurement locations. Given that the feature scale / identifiability relationship has never been quantified in any scenarios at all, we believe that choosing a common flow scenario as a basis for our analyses is a good choice.

We do take the point that it is possible that the relations we have uncovered may in some sense specific to the assumptions we have made, and realize that it is important to discuss the possible generality of the results. **We discuss the possible generality of the results in a new subsection, 4.4**.

> As a minor point, there was an earlier benchmarking of inverse models not mentioned in the opening, which addressed a tougher problem than the one discussed here published in Water Resources Research by Zimmerman et al. In 1988.

We appreciate that the reviewer brought this systematic inversion technique benchmarking paper to our attention. **We now mention the Zimmerman paper in the introduction as an example of the few papers performing rigorous comparison of inversion approaches**, preceding the works of Franssen et al. (2009) and Illman et al. (2010).

**Community comment**

> General comments
>
> Good theoretical research with implication on groundwater flow modelling and the engineering of the reservoirs where the geological flow heterogeneities are of paramount importance. Please, follow my guidance to improve the manuscript.

We appreciate the commenter's positive overall assessment.

> Specific comments
>
> Line 6. I suggest "This technique allows unbiased". Add the word "technique".

**We have done this.**

> Lines 20-22. Mini-permeameter, slug, packer and pumping tests can be also used to identify flow heterogeneities and determine the hydraulic conductivity. Specify this point.
>
> Line 22. "Point-to-point tracer tests" to detect flow heterogeneities. Please, add recent literature on the topic:
>
> - Deleu, R., Frazao, S. S., Poulain, A., Rochez, G., & Hallet, V. (2021). Tracer Dispersion through Karst Conduit: Assessment of Small-Scale Heterogeneity by Multi-Point Tracer Test and CFD Modeling. Hydrology, ((4)
>
> - Lorenzi, V., Banzato, F., Barberio, M. D., Goeppert, N., Goldscheider, N., Gori, F., Lacchini A., Manetta M., Medici G., Rusi S., Petitta, M. (2024). Tracking flowpaths in a complex karst system through tracer test and hydrogeochemical monitoring: Implications for groundwater protection (Gran Sasso, Italy). Heliyon, 10(2)

– Poulain, A., Rochez, G., Van Roy, J.P., Dewaide, L., Hallet, V. and De Sadelaer, G., 2017. A compact field fluorometer and its application to dye tracing in karst environments. Hydrogeology Journal, 25

Lines 48–88. The literature on the topic is much broader.

[....]

Line 514. Please, add relevant literature on the topic.

The commmenter suggested substantially increasing the literature review. Realistically, the literature somewhat related to our topic is so broad that we could plausibly have surveyed hundreds or thousands of papers. We have kept the bibliography largely as it was, save for adding two references in response to remarks by Reviewers 1 and 2. The suggested literature on karst terrain in particular is quite far afield: we discuss heterogeneous, but continuous permeability fields exclusively in our manuscript.

Line 84. Disclose the 3 to 4 specific objectives of your research by using numbers (e.g., i, ii and iii).

We agree it is useful to outline the objectives around here. In response to this, and to remarks from the reviewers, **we have added a new paragraph at the start of subsection 1.3 enumerating our specific high-level objectives.**

Line 90. "Feature scale". This expression is difficult to understand. Do you mean "observation scale"?

We see that this was not clear in the original draft. **We now define what we mean by the term in the first sentence of the introduction of the revised paper.**

Line 199. "theoretical observations". Can you re-call the key equations instead?

**We now reference the key equations explicitly here.**

Line 501. "Other geophysical fields". (i) remind to the reader that the principal implications are in the calibration of groundwater flow models, (ii) which other implications/applications in geophysics? You can look at my general comments.

Regarding (i) **we now explicitly recall the fields studied.** Regarding (ii), we feel that it is best to leave the possible applications outside our field of study general.

Figures and tables

Figure 1. Very nice figure, but it needs some changes. (i) Make the rectangles closer, (ii) make words and numbers larger, and (iii) thicker the black nodes.

Figures 2-4. Make words and figures larger on x and y axes.

We believe the figure labels are large enough that they should be legible even if a figure is shrunk to single-column size; we have used the same formatting previously in other articles and it has rendered well in the published

version. In terms of other stylistic comments, we think the figures are effective and readable as they are. Thus, we have not altered the figures, except for revised Figure 1 in response to Reviewer 1.

---

## Editor Decision (ED1)

Even if a copy-editing of the manuscript (spelling, grammar, sentence structure) is mandatory and will be done automatically and even if English is not my mother tongue, I take the freedom of suggesting possible changes, which, together with other technical comments, will hopefully guarantee a good quality of the published paper.

1) Throughout the whole paper, words "period" and "frequency" are used. But the work consider spatial variations of the physical quantities, so that I would use "wavelength" and "wavenumber", instead.

2) Throughout the whole paper "small scale" is used. I know that "small/big scale" is often used in the scientific literature to denote small or big scale-lengths, but this is wrong, in my opinion. Think to geographical maps. A map at scale 1:1,000,000=$10^{-6}$ does not show many details: topographic maps at scale 1:10,000=$10^{-4}$ (i.e., 100 times greater!) provides many more details. Therefore, I prefer to use "fine-" vs "large-" scale or "short" vs "long" scale-length. In several cases, "scale length" should be preferred to "scale".

3) Line 4. Should "of" be erased?

4) Line 181. I wouldn't use "Fourier series" here, because Fourier series refers explicitly to the use of complex exponentials or sinusoidal functions, whereas here the authors refer to the linear combination of a generic orthogonal basis.

5) Figure 1. It is important to state whether the color maps for corresponding figures (lnK or h) in (a) and (b) share the same limits or if different limits are considered for figures referred to the same quantity.

6) Line 222. Should "calibration" be substituted with "forward modeling"?

7) Line 226. I would prefer "$(x_i,y_i)$", instead of "$<x_i,y_i>$", because $< >$ is used in (47) to denote an average and $< , >$ is often used in mathematics to denote inner products.

8) Line 237. Check expression "the whole number multiples". May be, "all the integer multiples"?

9) Line 261. Check "fields represented as". May be, "fields are represented by"?

10) Lines 261 & 262. Verb "to represent" is used three times in short distance.

11) Line 165. Expression "functions of corresponding" should be corrected.

12) Lines 175ff. "Pr" is used to denote a "probability density function", but it remembers "probability". A different symbol should be preferred.

13) Line 304. The use of Gaussian distributions is common, mostly because it permits to perform analytical computations and to simplify mathematical development. I wouldn't introduce this assumption as a matter of fact or as a property of general validity.

14) Line 342. "To" should be added after "corresponding", shouldn't it?

15) Line 350. Is "made" the right word? May be, "extracted", or something similar?

16) Line 361. Statement "iterations were counted only when there was a reduction in the loss function" sounds very strange to me. The steepest descent method should yield a reduction of the objective function at each iteration. To my knowledge, the best way to apply it in inverse problems for groundwater hydrology is to compute the direction of change according to the gradient of the objective function with respect to the parameters to be fitted and then to perform a 1-D minimization of the objective function along that direction. May be, here a fixed step along the direction opposite to the gradient was used at each iteration?

17)  Line 423. "For" should be written with upper case "F".
18)  Line 434. Check expression "spanned the K-L basis functions". I think that "by" is missing after "spanned".
19)  Line 438. Sentence "In the base case (featuring all available information) was shared with the main MC study" should be rephrased.
20)  Line 458. Acronym "iid" hasn't been defined, has it?
21)  Line 473. Is "that" missing after "idea"?

---

## Author Response (AR2)

**Response to second-round comments**

In this document we reproduce the editor and reviewer comments in gray boxes , with our responses following, explaining our reasoning. Explanation of how specifically we have altered the MS in response to the comments appears in **bold**.

**Editor's remarks**

> Both reviewers provide a positive assessment of the revised version and require a minor revision. The revised paper will be sent to the reviewers for a final check.

We have attempted to address the remaining reviewer remarks.

**Reviewer 1: Roseanna Neupauer**

> [V]ery minor editorial changes.

We are gratified that we seem to have mostly satisfied this reviewer. While specific editorial changes were not specified, we have passed over the document carefully and **corrected a few minor defects throughout the text**.

**Reviewer 2: Philippe Ackerer**

> The paper addresses an interesting issue, not new but not solved until now. It is a paper dedicated to a new methodology using a parameterization based on Karhunen-Loève expansions. The methodology is tested with a synthetic 'model aquifer' under simplified conditions. Using synthetic data set avoids wrong interpretation of the methodology performance due to uncertainty and measurement errors that are unavoidable when using real test cases. Of course, it questioned the feasibility of the method, but this is another issue.
>
> The paper is well written, the methodology well described and the results convincingly discussed.

We appreciate the overall positive assessment.

> Some comments that can be discussed:
>
> - Since the exact heads and hydraulic conductivities are known over the all domain, it may be interesting to analyze the interactions between head and hydraulic conductivities (using a cross variogram for example). This information could be included in the discussion about the spacing/density of the measurements.

We agree this is an interesting interaction to consider. We are presently working on a follow-up study in which we examine the added value of spatial correlation information, including the sort of cross-covariance data used in cokriging approaches. We hope that it will not be a problem to save this particular analysis for that manuscript.

> – I was surprised by the use of measured velocities. They cannot be measured, to my knowledge. In wells, flow rates are measured and the width of the captured zone is required to estimate the velocity. In the field, at least 3 piezometers are necessary to estimate the head gradient, but hydraulic conductivity is needed to compute an average velocity.

We are careful not to assume knowledge of the direction of the velocity vector in our cost function (3), only potentially the *magnitude* of the Darcy velocity, which is the quantity represented by $\tilde{q}$. This may be obtained passively at a monitoring well by use of, e.g., a point dilution test. We mention this on line 35 of the MS.

> – The head sampling is based on a regular grid. I do not see the interest of the regular grid for a heterogeneous aquifer.

The use of a regular grid is naturally an idealization. Our working hypothesis is that effective measurement density, not detailed well configuration, controls the threshold of feature identification. On this view, use of a regular geometry is harmless, and it simplifies analysis because there is an obvious single measurement spacing scale, rather than a distribution. We could also have run the calibration trials by randomly distributing fixed numbers of measurement locations throughout the domain, and do not think it would appreciably change our average results. It would, however, have increased complexity of the analysis, as there would generally be some calibration trials featuring tight, redundant well clusters that poorly sample the domain, for which the effective number of distinct sampling locations is significantly less than the notional number of wells. **We mention our reasoning in section 3.4 of the revised manuscript.**

> – Only 50 random fields are used. This is very low compared to the variance of the LnK and no reliable statistics can be drawn with such a number. What is the interest of these 50? Why not testing random fields with other properties (smaller LnK variance, lower integral scales, …).

We stress that we used *different sets* of 50 randomly-generated fields *for each* of the eight distinct calibration ensembles that we present in Figures 3 and 4, as well as the regularization-by-truncation trial summarized in Table 1: 450 in total. The same patterns linking measurement-scale-normalized feature scale, $T$, to average degree of improvement in identification at that scale due to calibration, $r$, were apparent in the various calibration ensembles for different $\Psi$ shown in these figures. This suggests that the patterns observed are not artifacts of a specific set of fields that were generated. While more simulations would naturally increase confidence in our results by reducing noise (i.e., vertical scattering in these figures), we do not believe they would alter the average trends we have identified. **In section 3.4 of the revised manuscript, we make this point explicitly, and also clarify that *different* fields were used in each calibration ensemble. We also correct the text to state that, for each ensemble presented, *exactly* 50 fields were used.**

We do agree that different heterogeneity statistics, and potentially even different correlation structures, would be interesting to consider with this same approach in follow-on research. However, there is a limit to how many parameters can feasibly be varied in a computationally-intensive study such as this one (which consumed thousands of compute hours, as is).

> – Optimization is stopped after 500 steps. Why? The stopping criterion should be based on the value of the objective function and/or it gradient. The number of required steps may be different, depending on the measurement density for example.

We observed that, for all Ψ considered, 500 iterations was sufficient to enter the "plateau" regime in which improvements to the objective function are not matched by significant further improvements in underlying $K$ field L2 reconstruction error (as seen in Figure 2): the maximum reductions in field reconstruction error have been approximately achieved. As we are quantifying achievable $K$ field reconstruction error, we considered it acceptable to terminate optimization at this point to manage computational cost, which was already very high for this study. In the absence of resource constraints, we would have done as recommended, optimizing all trials to fixed convergence criteria defined on the objective function. **We now mention our reason for imposing this fixed cutoff in section 3.4 of the revised manuscript.**

> Typo Line 302, 411

We thank the reviewer for bringing these to our attention. **These have now been corrected.**

---

## Author Response (AR3)

We have edited the accepted manuscript in response to the feedback. We appreciate the attention to detail. We generally rephrased where suggested, even in a few instances where we believe our existing text was unproblematic. Two points require further discussion:

**Regarding remark (2):** according to the Oxford English Dictionary, one definition of *scale* is "[r]elative or proportionate size or extent; degree, proportion". We consistently use the word *scale* in this sense when we speak of, for example, *feature scale,* or *large-scale features*, with implied proportionality of feature area to the characteristic (Voronoi) area associated with each monitoring well in the grid. This meaning of *scale* (dimension of real entity relative to characteristic length) is a related, but slightly different, meaning to the one discussed in the feedback (dimension of map or model relative to dimension of real entity). It is maybe regrettable that the dimension of the real entity is in the numerator in one sense and the denominator in the other, but this is a quirk of English. Our usage is correct according to an authoritative dictionary. Our usage is also, as acknowledged, in keeping with common practice in the geosciences.

In places where there might be ambiguity as to whether *small-scale* referred to short wavelength or low resolution, we rephrased. In other places, we believe keeping our existing terminology is the best decision for clarity and correctness.

**Regarding remark (18):** our description is correct as written. Our algorithm is essentially as described in the feedback, except we employ a dynamic step size. On every iteration, the direction of steepest descent is identified, and a step slightly larger than the last successful step is attempted in the downgradient direction. If the attempted step is too large, goes past the local minimum, and fails to reduce the objective function, the attempted step size is continually halved until the objective function is successfully reduced. A gradient-descent iteration is only considered completed once the objective function has been successfully reduced by means of a step downgradient. This procedure is described in the manuscript.